# Effects of Types of Horticultural Activity on the Physical and Mental State of Elderly Individuals

**DOI:** 10.3390/ijerph17145225

**Published:** 2020-07-20

**Authors:** Pei-Chun Tu, Wen-Chen Cheng, Ping-Cheng Hou, Yu-Sen Chang

**Affiliations:** 1Department of Horticulture and Landscape Architecture, National Taiwan University, Taipei 106, Taiwan; d05628004@ntu.edu.tw (P.-C.T.); r00628119@ntu.edu.tw (P.-C.H.); 2School of Nursing, Fooyin University, Kaohsiung 831, Taiwan; ns135@fy.edu.tw

**Keywords:** healthy horticulture, heart rate variability, salivary amylase activity, Profile of Mood States, delay aging, healthy aging, elderly

## Abstract

This study aimed to evaluate the effect of types of horticultural activities among elderly individuals in a senior citizen learning camp. We arranged four horticultural activities: Grass Doll, Kokedama, Rocky Leaf Prints, and Herb Tasting and Smelling. Twenty-seven senior citizens (60 to 76 years) were recruited to participate. We assessed their physiological and mental state before and after the activities by measuring heart rate variability (HRV), pulse rate, blood pressure, salivary amylase activity (SAA), and Profile of Mood States (POMS). Results showed that SAA was significantly lower after the Rocky Leaf Prints, Herb Tasting and Smelling, and Kokedama. Pulse rate had a downward trend after the four horticultural activities. The ratio of low frequency (LF) over high frequency (HF) results showed that the Rocky Leaf Prints had a larger downward trend, followed by Herb Tasting and Smelling. POMS scores indicated significant improvement for Rocky Leaf Prints and Herb Tasting and Smelling. The types including artistic creation and food tasting were promising methods for psychological relaxation, stress reduction, and vigor enhancement for elderly persons. Findings showed that the impact of activities involving contact with plants was more significant in short-term activities. We hope this study can help plan the horticultural activities of the elderly in the future.

## 1. Introduction

Considering the advancement of medical technology and declining birthrates, an aging population structure society has become a widespread phenomenon in developed countries around the world. The issues faced by an aged society are getting more attention from the government and the population. Taiwan became an aged society in March 2018 [1]. Each individual’s cause and degree of aging is different, and aging includes functional aging, psychological aging, pathological aging, and social aging. According to a report in 2015, by the Ministry of Health and Welfare of Taiwan, the healthy or subhealthy accounted for 83.5% of the elderly population in Taiwan and the disabled account for 16.5% [2]. Whether healthy or disabled, as the elderly population ages they must face the decline of physiological function and cognitive function, which leads to the negative impact of their physiological, psychological, and social functions [3,4,5,6]. Studies have indicated that elders who contact nature or plants or engage in horticultural activities could maintain health, delay aging, and improve the benefits of physical, psychological, cognitive, and social functions [3,5,7,8,9,10,11,12,13,14,15,16]. Horticultural activities can also be used as a method for rehabilitation or an adjuvant treatment of chronic diseases. In addition, horticultural activities are popular and common leisure activities for the elderly [3,9,17,18,19,20,21]. One study revealed that investing in the development of disease prevention can save significant costs, as each person spends $10 per year and the rate of return is 5.6 times after five years [22]. The national health expenditure spending for chronic disease and long-term nursing institutions increased 1.23 times between 2015 and 2018 in Taiwan [23]. It is perceivable that prevention is better than a cure. Perhaps delaying aging through healthy horticultural activities can improve health as well as help avoid the cost of medical and long-term care [12,24].

There are many types of horticultural activities, including static and dynamic, indoor and outdoor, and touching plants. Horticultural activities are often used as auxiliary treatment methods particularly [9,21,25,26] because they stimulate the five senses [18]. The current general horticultural therapeutic activity types are summarized as follows: cultivation technique, artistic creation, food tasting or smelling, flower viewing, guided tour, making daily necessities, and so on [4,7,12,15,17,21,26,27,28]. Each type of activity has different benefits and should be used as an improvement policy for different objects with suitable activities. In addition to the types of activities, the activity field is also extremely important. Studies revealed that the indoor environmental factors, such as air quality and temperature, affect learning and health [29,30,31,32]. Therefore, the quality of the environment must also be considered during the event.

Many studies have confirmed that heart rate variability (HRV), blood pressure, pulse rate, and cortisol values can be measured before and after an activity to understand the benefits of the activity on the participants [28,33,34,35]. Psychological stress affects the body in two main ways: the hypothalamus-pituitary-adrenal gland axis and the sympathetic nervous system [36]. When people feel stress, the sympathetic nervous system becomes more active, which is related to increased heart rate. When this stress disappears, the parasympathetic nervous system decreases the heart rate [37]. The response of the autonomic nervous system can be monitored by the HRV. Therefore, HRV may be used as an objective assessment of stress and mental health [36]. HRV parameters include high frequency (HF), low frequency (LF), and the standard deviation of the normal-to-normal (NN) intervals (SDNN). The HF power is mainly mediated by parasympathetic nerves; the LF power is mainly mediated by sympathetic nerves [36,37]. The LF value decreases and the HF value increases when relaxed. It can also be evaluated by the ratio of the LF over the HF (LF/HF) [38]. SDNN is a measure of the total power in the analyzed segment [35,39]. When the SDNN value increases, it is an index of physiological resilience against stress [36]. The pulse and blood pressure tend to slow down as the mood relaxes. In addition, cortisol levels can represent the mental and physical states [4,40], however, the analysis time is longer and not instantaneous. One study confirmed that salivary amylase activity (SAA) is positively related to it and is used to express the current stress index. When the mood is nervous or more stressed, the SAA value is increased [40,41]. Moreover, we can use questionnaires or scales to understand more subjective mental states [16,42]. Commonly used questionnaires are the Profile of Mood States (POMS) [33,34,43], Life Satisfaction Index [5,16,44], stress scale [28], and depression related scale [6,15,18,24]. Many studies have explored how contact with nature and engagement in horticultural activities affects mental states [5,6,7,16,33,34,42]. Results have shown the positive emotions increased and the negative emotions decreased for five types of people. First, the elders living in nursing homes who were cognitively intact, and no visual or auditory impairment occurred to them [5,7,16]. Second, the elders with depressive symptoms and memory problems [6]. Third, middle-aged and elderly individuals with or without chronic diseases [34]. Fourth, those with cardiovascular disease [42]. Finally, the elders [44] and adults [33] without diseases.

Finally, whether it is horticultural therapy for people with diseases or horticultural activities for healthy or subhealthy people, many studies have confirmed that these activities can effectively improve physical and mental health, increase well-being, and achieve the effect of delaying aging [7,8,17,18,26,42]. However, few studies have explored the impact of activity types on the elderly. Therefore, this study selected three types of horticultural activities (culture technique, artistic creation, and food tasting) to measure blood pressure, pulse rate, SAA, and questionnaire responses before and after each activity. HRV was monitored throughout the process to understand the status of the elderly during activities. The results of this study can be used to plan healthy horticultural activities for the elderly in the future to delay aging, increase healthy and active aging, and increase happy learning.

## 2. Materials and Methods

### 2.1. Participants

The participants were 27 elders aged 60 to 76 years (average years 67.9 ± 4.5) who attended Fooyin University Senior Citizens Learning Camp in Kaohsiung, Taiwan. The Senior Citizens Learning Camp is similar to the third age university (U3A), to encourage the senior learning motivation and enhance their physical and mental health [45,46]. Taiwan’s Ministry of Education (MOE) has promoted the senior education since 2008. This project comprised of universities and colleges that set up the Senior Citizens Learning Camp, which was conducted at the school. The major group of the elderly in these universities is citizens over 55 years old. Moreover, they are healthy or subhealthy and can act independently without support or care. For recruitment in this study, program details including inclusion criteria (i.e., elderly persons who were not disabled and can act independently) were posted on a bulletin board at Fooyin University Senior Citizens Learning Camp. The 27 participants included 5 men and 22 women and they participated in five activities. First there was nonhorticultural intervention, taken as the baseline. It was conducted on 4 January 2019. Participants conducted the most common leisure activity for the elderly in Taiwan, which was watching TV [47]. The other four times were with horticultural intervention. This study was approved by the Research Ethics Committee of National Taiwan University (NTU-REC No. 201807HM036).

### 2.2. Healthy Horticultural Program

The program was conducted twice a week from January 8 to 18 January 2019, for four sessions (average 1 h/session). During each healthy horticultural session, there were four phases (Figure 1): (1) Pretest (5 min); (2) Activity content explanation (10 min); (3) Healthy horticultural activity (40 min); (4) Post-test (5 min). Participants performed healthy horticultural sessions in groups of five to six people. The session types and titles of the four-session horticultural activity program are listed in Table 1. The first two sessions focused on culture technique. In Session 1, Grass Doll was an activity based on the skills of sowing and seed germination. In Session 2, Kokedama was an activity about potting, transplanting, and culture technique. In Session 3, Rocky Leaf Prints was an activity based on artistic creation combined with hand massage. In Session 4, Herb Tasting and Smelling was an activity that combined smell and taste to learn about different herbs (e.g., mint, rosemary, lavender, and stevia).

### 2.3. Physiological Indices

To assess the objective physiological state of the program before and after the five activities, HRV, pulse rate, blood pressure (systolic and diastolic), and salivary amylase activity (SAA) were used to evaluate autonomic nervous system activity. HRV was measured throughout the events using an equipment electrocardiography (ECG) (ECG125, BeneGear Inc., New Taipei City, Taiwan). This model contacted Wi-Fi (wireless Internet) and enabled transmission of the measured data to the computer. The HRV parameters included LF, HF, LF/HF, and SDNN. We had the LF and HF component as a measure of sympathetic and parasympathetic nervous system activity of HRV. SDNN represented total variability, and higher values indicated increased parasympathetic activity. Pulse rate and blood pressure were measured using a fully automatic wrist type digital blood pressure monitor (HEM-6121, Omron Healthcare Co Ltd., Matsusaka, Japan). We measured SAA value (kIU/L) with a salivary amylase monitor (CM-21, Nipro, Osaka, Japan) to evaluate the level of stress [40,48]. To determine the effects of the five activities and eliminate individual differences, we interpreted the changes in the LF/HF, SDNN, blood pressure, pulse rate, and SAA for each activity by dividing the post-test results by the pretest results (the ratio of phase 4/phase 1).

### 2.4. Psychological Indices

To assess the subjective emotional condition of the participants before and after these activities, a short form of the POMS subscale scores and a Total Mood Disturbance (TMD) score were used to evaluate psychological responses to the horticultural activities. The POMS scores with 37 questions using the five-point Likert Scale were determined for the following six subscales, including depression (D), confusion (C), tension (T), anger (A), fatigue (F), and vigor (V). The TMD score was calculated by summing the subscales of depression (D), confusion (C), tension (T), anger (A), and fatigue (F) and subtracting the subscales of vigor (V) (TMD = D + C + T + A + F − V). A high TMD score indicated a poor mental state, and a low score indicated a better mental state [33].

### 2.5. Environment Indices

Studies have indicated that indoor air quality affects people’s work efficiency and judgment ability [30,32,49]. In order to ensure the environmental quality of the operation space, the parameters of temperature, relative humidity, concentration of carbon dioxide (CO_2_), and concentration of carbon monoxide (CO) were monitored by AIRBOXX Indoor Air Quality Monitor (KD Engineering, NY, USA). We started recording 30 min before each activity and recorded data at 10-minute intervals until 30 min after the events ended. The environmental space was about 98 square meters, and three monitoring points were placed in a classroom. The temperature, relative humidity, light intensity, concentration of CO_2_, and concentration of CO in the air were 23.6 ± 0.7 °C (mean ± standard deviation), 83.8 ± 1.4%, 890.0 ± 76.9 lux, 803.9 ± 233.0 ppm, and 1.7 ± 0.5 ppm, respectively.

### 2.6. Statistical Analysis

The data were analyzed using IBM SPSS Statistics version 26.0 (IBM Corp., Armonk, NY, USA) and CoStat 6.2 (CoHort Software, Monterey, CA, USA), and the results were plotted by SigmaPlot software version 10.0 (Systat Software Inc., San Jose, CA, USA). We used paired sample *t*-tests to compare the difference between the pretest and post-test at *p* < 0.05 was considered statistically significant. Each activity was evaluated for significance using an analysis of variance (ANOVA) followed by a least significant difference (LSD) test at *p* < 0.05. The effect size was reported by Cohen’s d.

## 3. Results

### 3.1. Psychological Indices

The results for SAA were significantly decreased after the activities of Rocky Leaf Prints (*p* < 0.05), Herb Tasting and Smelling (*p* < 0.05), and Kokedama (*p* < 0.001). The results for pulse rate had a downward trend after the four healthy horticultural activities. In addition, the results for systolic and diastolic blood pressure had an upward trend but did not significantly change after the four healthy horticultural activities (Table 2, Table 3, Table 4 and Table 5).

Table 6 showed the changes in LF/HF, SDNN, blood pressure, pulse rate, and SAA before and after activity (the ratio of phase 4 to phase 1) for these activities. There was a downward trend in LF/HF, pulse rate, and SAA (<1 indicates a decline) in these four horticultural activities. Their decreased level was similar. Herb Tasting and Smelling and Rocky Leaf Prints decreased more, followed by Kokedama and Grass Doll. There was an upward trend in SDNN and blood pressure (>1 indicates an increase), but there was no statistical difference between these activities.

The results of the changes in participants’ LF/HF and SDNN in the four phases for the four healthy horticultural activities are shown in Figure 2. The results indicated that LF/HF was the highest during the pretest (phase 1). The results showed a downward trend in phase 2, followed by a slight increase in activity operations, and then a downward trend in the post-test (phase 4). The results for SDNN had an upward trend during phase 3 to phase 4.

### 3.2. Physiological Indices

The results for the POMS scores indicated improvement after the four healthy horticultural activities. For the Grass Doll activity, four of the negative mood states of the POMS, “depression”, “confusion”, “tension”, and “fatigue” significantly decreased (*p* < 0.05) from pretest to post-test (Table 7). For the program of Kokedama, “anger” of the negative subscales of the POMS significantly decreased (*p* < 0.05) and the “vigor” of the positive subscale of the POMS significantly increased (*p* < 0.01) from pretest to post-test (Table 8). All the POMS scores indicated significant improvement for the activities of Rocky Leaf Prints (Table 9) and Herb Tasting and Smelling (Table 10). Five of the negative mood states of the POMS significantly decreased (*p* < 0.01) and the positive mood state of the POMS significantly increased (*p* < 0.001) from pretest to post-test. Results for total mood disturbance (TMD) in POMS had a downward trend after the four healthy horticultural activities, Herb Tasting and Smelling and Rocky Leaf Prints decreased more, followed by Kokedama and Grass Doll (Table 7, Table 8, Table 9, and Table 10 and Figure 3).

## 4. Discussion

This study compared the physiological and psychological responses to the types of horticultural activities among elderly individuals. Pulse rate and SAA were determined to be lower after every activity. Previous studies indicated that when the mood is relaxed, the pulse rate is decreased [34,42]. SAA represented the current state of mental stress and physical fatigue because these measurement values decrease when a person is relaxed [40,48,50]. Considering this, the four horticultural activities in this study had a relaxing effect on the elderly. Moreover, previous studies showed that blood pressure is significantly lower in the horticultural therapy program [27,51]. However, systolic and diastolic blood pressure were not significantly lower after every activity in this study. It might be that previous studies were mostly long-term programs which could explain the tendency for the blood pressure to decrease [16,27,51]. This study mainly focused on relatively short-term activities, so there was no significant decrease in blood pressure. The results of Table 6 showed that the effects of five activities on the changes of physiological indices, the four horticultural activities’ SAA, pulse rate, and LF/HF decreased significantly. It showed that it was relaxable and pressure relievable. Previous studies indicated that SAA, pulse rate, and LF/HF can represent immediate emotion and pressure [38,40]. This is the reason why SAA, pulse rate, and LF/HF are used for the level of pressure relieving of short-term horticultural activities, which is consistent with the findings of previous studies [33,34,50]. Considering the changes in SAA, pulse rate, and LF/HF in this study, we found that Rocky Leaf Prints (artistic creation) and Herb Tasting and Smelling (food tasting) were better to relieve the stress and anxiety of the elderly.

HRV measurements have also been used to evaluate emotional and health-related physiological effects [33,37,39,52]. The ratio of LF/HF can be used to represent the change of sympathetic and parasympathetic nerves. When the ratio of LF/HF decreases, it indicates physiological relaxation [33,52]. Higher SDNN reflect a lower mental effort [53]. In our study, LF/HF decreased and SDNN increased after participating in the four horticultural activities. The ECG was worn throughout this study, so we interpreted the changes in LF/HF and SDNN in the four phases (pretest, activity content explanation, healthy horticultural activity, and post-test) of the activity. During the process of these four horticultural activities, there was a downward trend in phase 2. This means that it was more relaxable. Perhaps the reason is related to the instructors’ teaching styles. Some studies indicated that students’ learning outcomes [54] and learning anxiety [55] were related the instructors’ teaching styles. To avoid the error of experiment caused by instructors, the same instructor was used in these four activities. The instructor explains the activities in such a way that testers pay attention and listen carefully. This made them calm down, which caused relaxable effects. In phase 3, due to the activities, there was an upward trend of LF/HF and SDNN. After joining four different types of horticultural activities, compared to the pretest, the downward range of LF/HF was caused by different effects of four activities. Observing the emotional and physiological changes of the elderly during the horticultural activities can be used as a plan for future activities for the elderly.

Previous studies showed that horticultural activities had mental benefits for the elderly [13,16,33,42], which is consistent with our findings. The POMS questionnaire indicated that the negative mood states were significantly lower after the four horticultural activities. Additionally, the positive mood state of vigor was significantly higher, and the total mood disturbance decreased significantly after the horticultural activities, except for Grass Doll. Therefore, we concluded that the types and materials of horticultural activities (with plants or without plants) affect the mental state of the elderly.

Regarding the changes in pulse rate, SAA, LF/HF, and POMS in this study, we found that artistic creation-type (Rocky Leaf Prints) and food tasting-type (Herb Tasting and Smelling) showed better effects in these activities. Maybe this is because they are able to create freely and experience immediately making their effects better. In addition, culture technique-type was better than the control group. However, the effects were less than types of artistic creation and food tasting. This is due to the long-term effects of the culture technique-type, such as taking care of plants and observing seed germination and the growth of plants. Previous studies were mostly in the plan of long-term horticultural activities [16,18,24,27], so there was a benefit in their studies about culture technique-type.

In the culture technique-type, Kokedama had a more relaxing effect than the Grass Doll. According to previous studies, contact with plants is more relaxing than contact without plants [33,49,56,57], which is the same as this study. Kokedama involved contact with plants during the operation, which may be the reason the effect was immediate. The Grass Doll was a sowing activity; therefore, its main benefit was in the follow-up observation of seed germination and continued activities. We concluded that for short-term horticultural activities, we can arrange the artistic creation and food tasting types to have as much contact with plants as possible.

Many studies proved that horticultural activities increase happiness and also improve physical and psychological status [8,18,25,26]. However, fewer studies discussed the benefits of different types of horticultural activities. Therefore, this study researched the effects of different types of horticultural activities. Hopefully, it can be used as the planning of the horticultural activities in the future. Horticultural activities are not only convenient in operating but also in material gathering. Besides, compared to forest healing [34], area limits do not exist. Activities can be conducted in senior citizens learning camp and long-term care facilities in cities. Nevertheless, there were still some limits. Firstly, the same instructor was used in these four horticultural activities. Besides, it is required for the instructors to be trained so that they are skillful, since the teaching style affects the effect. Second, the POMS scale had many questions and took a long time to answer since the object of this study was elders. Hopefully, a simplified and reliable scale can be obtained to evaluate the effects of activities for elders. Third, in evaluating physical and psychological status, we did not observe significant changes in the blood pressure and SDNN. Previous studies are mainly about the discussion about the effects of joining long-term horticultural activities [16,24,27]. Therefore, blood pressure can be used as the assessment of long-term horticultural activities; however, it was unable to indicate the results immediately. Besides, this study can evaluate the items of effects on short-term activities. SAA, LF/HF, pulse rate, and POMS are included. Hopefully, we can understand how different types of horticultural activities benefit each age range by using this method and promote learning effects through different types of planning about horticultural activities. This can be used for activity planning in the future, achieving the benefits of healing and pressure relieving while also promoting social well-being.

## 5. Conclusions

This study measured the pulse rate, SAA, HRV, and POMS scales to understand the effects of horticultural activities, as well as the benefits of three types of horticultural activities, culture techniques, artistic creation, and food tasting, for the elderly. The results showed that joining horticultural activities were better than baseline. Besides, artistic creation and food tasting types were better than culture techniques in horticultural activities. Additionally, in short-term horticultural activities, the benefits of activities in contact with plants were immediate. We hope that this study can help plan the horticultural activities of the elderly, which can improve well-being and help achieve active aging.

## Figures and Tables

**Figure 1 ijerph-17-05225-f001:**
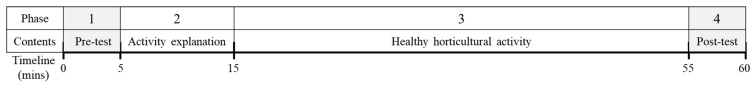
The operation process and time of the activity.

**Figure 2 ijerph-17-05225-f002:**
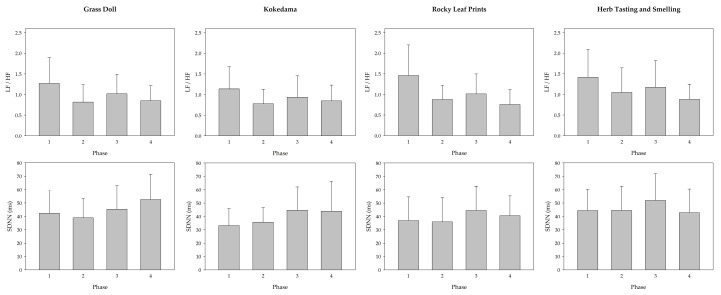
Effect of various types of horticultural activities on the changes in the ratio of the low frequency over high frequency (LF/HF) and the standard deviation of the NN intervals (SDNN). N = 27, mean ± standard deviation.

**Figure 3 ijerph-17-05225-f003:**
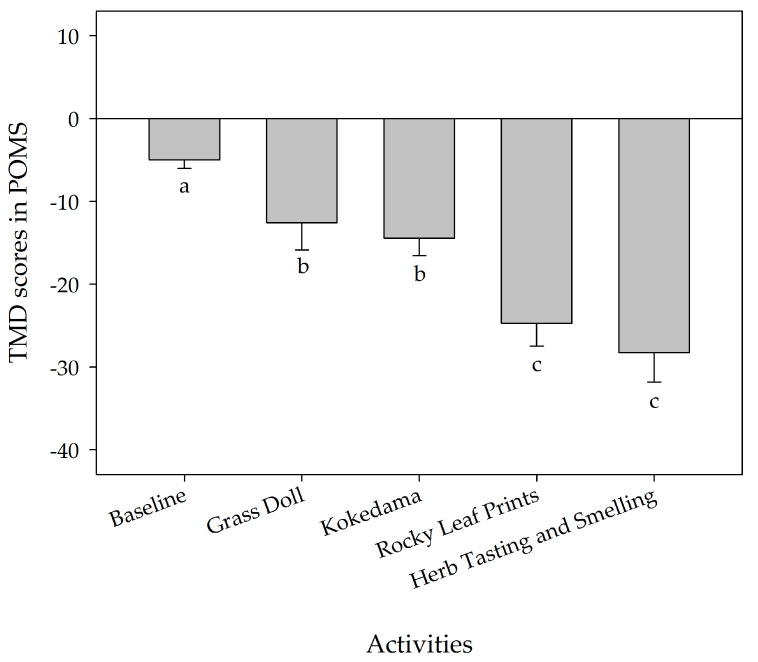
Effect of various types of horticultural activities on the total mood disturbance (TMD) in the Profile of Mood State (POMS). N = 27, mean ± standard error. ^a, b, c^ Means within a treatment followed by different letters significantly differ at *p* ≤ 0.05 by the least significant difference (LSD) test.

**Table 1 ijerph-17-05225-t001:** Activity titles, types, and objectives of the healthy horticultural activity program.

Activity Title	Types	Method	Objectives
Grass Doll	culture technique	Using socks, put seeds and peat soil into a doll style	To learn the skills of sowing and seed germination. To understand the process of seed germination and experience the growth process of life.
Kokedama	culture technique	Wrap the plant in moss and tie it with cotton thread	To learn the skills of potting, transplanting and culture technique. To train the ability of hand grip and coordination.
Rocky Leaf Prints	artistic creation	Using stone balls to print the color of leaves and flowers onto cotton cloth	To understand that plant color changes are affected by plant pigments and the environment. To be creative and artistic.
Herb Tasting and Smelling	food tasting	Smell and taste to learn about different herbs	To learn the knowledge and application of herbal plants. To enjoy the experience of smell and taste.

**Table 2 ijerph-17-05225-t002:** Effect of Grass Doll on physiological indices.

Physiological Indices	Pre-Test	Post-Test	*p*	*t*	Effect Size
Systolic blood pressure (mmHg)	136.6 ± 4.0	138.0 ± 2.8	0.771	0.292	0.08
Diastolic blood pressure (mmHg)	80.5 ± 2.1	81.7 ± 1.9	0.676	0.421	0.12
Pulse rate (bpm)	75.7 ± 2.2	72.0 ± 2.2	0.234	−1.206	0.35
Salivary amylase activity (KU/L)	32.1 ± 6.5	17.3 ± 3.6	0.053	−1.985	0.57

Data are expressed as the mean± standard error. *t*: The *t*-test statistic for a paired sample *t*-test; *p*: The *p*-value (probability value) for the *t*-test statistic. The effect size is reported by Cohen’s d.

**Table 3 ijerph-17-05225-t003:** Effect of Kokedama on physiological indices.

Physiological Indices	Pre-Test	Post-Test	*p*	*t*	Effect Size
Systolic blood pressure (mmHg)	131.5 ± 3.7	136.5 ± 3.7	0.359	0.927	0.28
Diastolic blood pressure (mmHg)	76.5 ± 3.7	81. 5 ± 1.8	0.253	1.160	0.35
Pulse rate (bpm)	74.2 ± 2.9	72.3 ± 2.9	0.650	−0.457	0.14
Salivary amylase activity (KU/L)	28.8 ± 3.4	14.5 ± 1.9	0.000 ***	−3.548	1.07

Data are expressed as the mean± standard error. *t*: The *t*-test statistic for a paired sample *t*-test; *p*: The *p*-value (probability value) for the *t*-test statistic. *** was considered significant at *p* < 0.001 paired-sample *t*-test. The effect size is reported by Cohen’s d.

**Table 4 ijerph-17-05225-t004:** Effect of Rocky Leaf Prints on physiological indices.

Physiological Indices	Pre-Test	Post-Test	*p*	*t*	Effect Size
Systolic blood pressure (mmHg)	130.4 ± 3.6	135.2 ± 3.7	0.361	0.922	0.27
Diastolic blood pressure (mmHg)	78.0 ± 2.2	79.3 ± 2.1	0.667	0.433	0.13
Pulse rate (bpm)	80.7 ± 2.8	75.3 ± 2.4	0.155	−1.446	0.43
Salivary amylase activity (KU/L)	58.0 ± 16.4	13.2 ± 2.6	0.012 *	−2.636	0.78

Data are expressed as the mean± standard error. *t*: The *t*-test statistic for a paired sample *t*-test; *p*: The *p*-value (probability value) for the *t*-test statistic. * was considered significant at *p* < 0.05 paired-sample *t*-test. The effect size is reported by Cohen’s d.

**Table 5 ijerph-17-05225-t005:** Effect of Herb Tasting and Smelling on physiological indices.

Physiological Indices	Pre-Test	Post-Test	*p*	*t*	Effect Size
Systolic blood pressure (mmHg)	133.6 ± 4.0	137.0 ± 4.0	0.571	0.571	0.17
Diastolic blood pressure (mmHg)	81.9 ± 2.1	83.5 ± 2.2	0.605	0.521	0.16
Pulse rate (bpm)	75.8 ± 2.3	70.0 ± 2.0	0.075	−1.825	0.55
Salivary amylase activity (KU/L)	57.4 ± 13.3	18.0 ± 4.2	0.010 *	−2.714	0.82

Data are expressed as the mean± standard error. *t*: The *t*-test statistic for a paired sample *t*-test; *p*: The *p*-value (probability value) for the *t*-test statistic. * was considered significant at *p* < 0.05 paired-sample *t*-test. The effect size is reported by Cohen’s d.

**Table 6 ijerph-17-05225-t006:** Effect of various types of horticultural activities on the changes of physiological indices.

Activities	Types	Phase 4 / Phase 1					
		LF/HF ^z^	SDNN ^z^	Blood Pressure		Pulse Rate	SAA ^z^
				Systolic	Diastolic		
Baseline	− ^x^	1.19 ± 0.42 a ^y^	1.10 ± 0.24 a ^y^	1.05 ± 0.16 a ^y^	1.06 ± 0.13 a ^y^	1.02 ± 0.08 a ^y^	1.10 ± 0.31 a ^y^
Grass Doll	culture technique	0.80 ± 0.44 bc	1.96 ± 3.52 a	1.02 ± 0.09 a	1.02 ± 0.06 a	0.95 ± 0.06 bc	0.79 ± 0.78 ab
Kokedama	culture technique	0.84 ± 0.41 b	1.45 ± 0.85 a	1.05 ± 0.15 a	1.48 ± 0.08 a	0.98 ± 0.08 b	0.56 ± 0.15 bc
Rocky Leaf Prints	artistic creation	0.58 ± 0.26 c	1.21 ± 0.38 a	1.04 ± 0.09 a	1.02 ± 0.09 a	0.94 ± 0.09 bc	0.50 ± 0.09 c
Herb Tasting and Smelling	food tasting	0.71 ± 0.32 bc	0.99 ± 0.28 a	1.03 ± 0.10 a	1.02 ± 0.05 a	0.92 ± 0.05 c	0.36 ± 0.10 c

^z^ Low frequency (LF), high frequency (HF), the standard deviation of the NN intervals (SDNN), and salivary amylase activity (SAA). ^y^ Means separation within columns followed by the different letter(s) are significantly different at *p* < 0.05 by the least significant difference (LSD) test. Data are expressed as the mean± standard deviation. ^x^ −: There was nonhorticultural intervention.

**Table 7 ijerph-17-05225-t007:** Effect of Grass Doll on Profile of Mood State (POMS).

Subscales	Pre-Test	Post-Test	*p*	*t*	Effect Size
depression-dejection	15.8 ± 0.8	13.5 ± 0.9	0.049 *	−2.024	0.58
confusion-bewilderment	12.4 ± 0.8	10.2 ± 0.7	0.046 *	−2.054	0.59
tension-anxiety	13.7 ± 1.0	11.0 ± 0.8	0.043 *	−2.077	0.60
anger-hostility	14.8 ± 1.1	12.3 ± 0.9	0.051	−1.750	0.51
fatigue-inertia	11.3 ± 0.9	9.3 ± 0.8	0.004 **	−2.978	0.51
vigor-activity	22.3 ± 0.8	23.0 ± 1.0	0.528	0.635	0.18
Total Mood Disturbance	45.8 ± 4.4	33.2 ± 4.3	0.046 *	−2.046	0.59

Data are expressed as the mean ± standard error. *t*: The *t*-test statistic for a paired sample *t*-test; *p*: The *p*-value (probability value) for the *t*-test statistic. * was considered significant at *p* < 0.05; ** was considered significant at *p* < 0.01 paired-sample *t*-test. The effect size is reported by Cohen’s d.

**Table 8 ijerph-17-05225-t008:** Effect of Kokedama on Profile of Mood State (POMS).

Subscales	Pre-Test	Post-Test	*p*	*t*	Effect Size
depression-dejection	16.0 ± 1.2	13.4 ± 0.9	0.083	−1.773	0.53
confusion-bewilderment	12.0 ± 0.7	10.3 ± 0.8	0.135	−1.522	0.46
tension-anxiety	13.3 ± 1.0	11.2 ± 0.9	0.138	−1.513	0.46
anger-hostility	14.6 ± 1.1	11.8 ± 0.8	0.049 *	−2.020	0.61
fatigue-inertia	11.0 ± 0.9	9.0 ± 0.7	0.098	−1.691	0.51
vigor-activity	21.0 ± 0.7	24.5 ± 0.7	0.001 **	3.492	1.05
Total Mood Disturbance	45.7 ± 4.9	31.3 ± 4.5	0.035 *	−2.176	0.66

Data are expressed as the mean ± standard error. *t*: The *t*-test statistic for a paired sample *t*-test; *p*: The *p*-value (probability value) for the *t*-test statistic. * was considered significant at *p* < 0.05; ** was considered significant at *p* < 0.01 paired-sample *t*-test. The effect size is reported by Cohen’s d.

**Table 9 ijerph-17-05225-t009:** Effect of Rocky Leaf Prints on Profile of Mood State (POMS).

Subscales	Pre-Test	Post-Test	*p*	*t*	Effect Size
depression-dejection	16.0 ± 1.1	12.6 ± 0.8	0.017 *	−2.484	0.73
confusion-bewilderment	11.4 ± 0.7	8.7 ± 0.6	0.004 **	−3.070	0.91
tension-anxiety	13.2 ± 0.9	9.2 ± 0.6	0.000 ***	−3.786	1.12
anger-hostility	14.7 ± 0.9	9.3 ± 0.5	0.000 ***	−5.275	1.56
fatigue-inertia	11.5 ± 0.6	7.6 ± 0.5	0.000 ***	−5.127	1.51
vigor-activity	21.2 ± 0.5	26.4 ± 0.5	0.000 ***	7.212	2.13
Total Mood Disturbance	45.6 ± 4.1	20.9 ± 3.2	0.000 ***	−4.763	1.40

Data are expressed as the mean± standard error. *t*: The *t*-test statistic for a paired sample *t*-test; *p*: The *p*-value (probability value) for the *t*-test statistic. * was considered significant at *p* < 0.05; ** was considered significant at *p* < 0.01; *** was considered significant at *p* < 0.001 for paired-sample *t*-test. The effect size is reported by Cohen’s d.

**Table 10 ijerph-17-05225-t010:** Effect of Herb Tasting and Smelling on Profile of Mood State (POMS).

Subscales	Pre-Test	Post-Test	*p*	*t*	Effect Size
depression-dejection	19.3 ± 1.8	12.7 ± 0.8	0.002 **	−3.382	1.02
confusion-bewilderment	11.6 ± 0.6	8.9 ± 0.6	0.004 **	−3.091	0.93
tension-anxiety	12.9 ± 0.6	8.6 ± 0.5	0.000 ***	−5.403	1.63
anger-hostility	14.9 ± 0.8	9.6 ± 0.6	0.000 ***	−5.016	1.51
fatigue-inertia	11.2 ± 0.6	7.1 ± 0.5	0.000 ***	−5.073	1.53
vigor-activity	20.7 ± 0.8	26.0 ± 0.5	0.000 ***	5.650	1.70
Total Mood Disturbance	49.2 ± 4.2	21.0 ± 2.8	0.000 ***	−5.580	1.68

Data are expressed as the mean± standard error. *t*: The *t*-test statistic for a paired sample *t*-test; *p*: The *p*-value (probability value) for the *t*-test statistic. ** was considered significant at *p* < 0.01; *** was considered significant at *p* < 0.001 paired-sample *t*-test. The effect size is reported by Cohen’s d.

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
