# Peer review of "Effects of Types of Horticultural Activity on the Physical and Mental State of Elderly Individuals"

_ijerph, 2020, doi:10.3390/ijerph17145225_

Round 1

Reviewer 1 Report

effects of types horticultural activity on the physical and mental state of elderly individuals.

Thank you for the opportunity to review this interesting manuscript, which reported the findings of a horticultural program with 4 activities on various measurements for elderly adults.

The major comment on this manuscript is:

in the Rocky Leaf Prints session, a hand massage is included. This is not a horticultural activity, and from the literature it has been shown to reduce stress levels on it’s own. this thus limits the ability to compare this session with the other sessions, and compromises your conclusions that creative and artistic components are successful.

Abstract:

POMS are not reported, however these were statistically significant. and are more relevant to the trends of the LF/HF results.

line 61-62:  this sentence needs clarification.

line 62-66:  consider use of the word power, this is unclear

line 71 – reference needed for POMS and stress scale.

line 73.-74  the final sentence needs expansion, and to clarify if these are for adults with disease (eg CVD), or healthy adults?

Methods

for those readers outside Taiwan, unfamiliar with what a Learning Camp. description would aid understanding, eg. are the adults attending this camp independent living, what is their health state, is this program specific to a health condition, eg. cardiac rehab.

explain where in the camp the sessions were held, eg building?

Results

there is a repetition in the text of results presented in the tables, which is not needed, in particular line 153-156, 193-207

Fig 3- present in order of manuscript. a,b explanation missing

Discussion

there is a lack of critique of previous studies in comparison to the current study, and explanations as to reasons why these may have differed or are similar.

line 236-237  this study has not proved that pulse rate and SAA can be used as an index. the authors have chosen to deduce that from these 2 measures that 2 of these activities relieve the stress and anxiety of elderly. 

Suggestions as to why this did not occur with the other 2 activities would aid this area.

need to include an explanation of SDNN effect

Lines 244-250 are not discussion of this study – thave not calculated this ratio nor comparison to previous studies.

in the abstract, state that artistic creation and food tasting are promising, however this is not addressed in the discussion.

lines 266-270  are repetitive.

Line 184-188 – this was not addressed in the discussion. these results show the importance of Phase 2 and being comfortable and relaxed, which can be affected by the rapport with the instructor, thus instructor and environment may be a key factor.

No strengths and limitations of this study are presented in the discussion.

conclusion

use of word gardening inconsistent with horticultural in rest of manuscript

Grammar throughout the article would benefit from proof reading.

Consistency in the order of activities in written text to match the order in the results section would aid readability.

Author Response

Dear Reviewer:

The manuscript has been rechecked and the necessary changes have been made in accordance with the reviewer’s suggestions. The responses to all comments have been prepared and attached. 

We thank you for your thoughtful suggestions and insights. The manuscript has benefited from these insightful suggestions. 

If there are any other questions, please let me know. Thank you for your time and kind assistance.

Reviewer 2 Report

This study aimed to estimate the effect of horticultural activities on biological and psychological states among elderly individuals. What the authors try to do is interesting.

However, I can't understand why they didn't put a control group in the experiment. In this kind of experiment, it seems that the control group is usually placed. They should have an argument to justify not having control. Otherwise, they should conduct an additional experiment to determine how biological and psychological measures change in the absence of any intervention. It is advisable to have the same 27 elderly people who participated in this experiment reassemble and observe how various indicators change in the absence of any intervention.

As there is no control group, the authors can say comparatively like “The results showed that the artistic creation and food tasting types are better than culture techniques” (in conclusion). However, they cannot say whether these activities were significantly beneficial to elderly people.

Moreover, some sort of test needs to be done to statistically say that one intervention is more effective than another, but it seems the authors did not.

In p.4 l.149, the authors say they use ANOVA. However, a series of tables give the impression that the T-test was performed.

By making additional experiments with a control group, improving the test method, and changing the abstract and conclusion part to a clearer format, it should be worth the publication.

Author Response

(The authors gave the same response as above.)

Round 2

Reviewer 2 Report

All the problems I have pointed out have been resolved.

Author Response

Thank you for your suggestions and comments to make the content more complete.